# GPT-Driver: Learning to Drive with GPT

## Abstract

We present a simple yet effective approach that can transform the OpenAI GPT-3.5 model into a reliable motion planner for autonomous vehicles. Motion planning is a core challenge in autonomous driving, aiming to plan a driving trajectory that is safe and comfortable. Existing motion planners predominantly leverage heuristic methods to forecast driving trajectories, yet these approaches demonstrate insufficient generalization capabilities in the face of novel and unseen driving scenarios. In this paper, we propose a novel approach to motion planning that capitalizes on the strong reasoning capabilities and generalization potential inherent to Large Language Models (LLMs). The fundamental insight of our approach is the reformulation of motion planning as a language modeling problem, a perspective not previously explored. Specifically, we represent the planner inputs and outputs as language tokens, and leverage the LLM to generate driving trajectories through a language description of coordinate positions. Furthermore, we propose a novel prompting-reasoning-finetuning strategy to stimulate the numerical reasoning potential of the LLM. With this strategy, the LLM can describe highly precise trajectory coordinates and also its internal decision-making process in natural language. We evaluate our approach on the large-scale nuScenes dataset, and extensive experiments substantiate the effectiveness, generalization ability, and interpretability of our GPT-based motion planner. Code will be released.

## 1 Introduction

Autonomous driving stands as one of the most ambitious and challenging frontiers in modern technology, aiming to revolutionize transportation systems globally. Central to this endeavor is the concept of motion planning, a cornerstone in autonomous driving technology that seeks to devise safe and comfortable driving trajectories for autonomous vehicles. The intricacies of motion planning arise from its need to accommodate diverse driving scenarios and make reasonable driving decisions. As autonomous vehicles interact with various environments and unpredictable human drivers, the robustness and explainability of motion planners become essential for driving safety and reliability.

Existing motion planning approaches generally fall into two categories. The rule-based methods (Treiber et al., 2000; Thrun et al., 2006; Bacha et al., 2008; Leonard et al., 2008; Urmson et al., 2008; Chen et al., 2015; Sauer et al., 2018; Fan et al., 2018) designed explicit rules to determine driving trajectories. These methods have clear interpretability but generally fail to handle extreme driving scenarios that are not covered by rules. Alternatively, the learning-based approaches (Bojarski et al., 2016; Codevilla et al., 2018; 2019; Rhinehart et al., 2019; Zeng et al., 2019; Sadat et al., 2020; Casas et al., 2021; Hu et al., 2022; 2023; Dauner et al., 2023) resorted to a data-driven strategy and learned their models from large-scale human driving trajectories. While exhibiting good performance, these approaches sacrifice interpretability by viewing motion planning as a black-box forecasting problem. Essentially, both prevailing rule-based and learning-based approaches are devoid of the common sense reasoning ability innate to human drivers, which restricts their capabilities in tackling long-tailed driving scenarios.

Recent advances in Large Language Models (LLMs) (Brown et al., 2020; Ouyang et al., 2022; OpenAI, 2023; Touvron et al., 2023a;b) have demonstrated great generalization power and common sense reasoning ability emerged from these language models, indicating their potential in addressing problems in the realm of autonomous driving. An important question naturally arises: How can we leverage LLMs to resolve the motion planning problem? The major challenge is that motion planners are required to process heterogeneous inputs, *e.g.*, ego-vehicle information, maps, and

perception results, and they need to predict high-precision waypoint coordinates that represent a future driving trajectory. While LLMs excel at language understanding and generation, they cannot directly handle these heterogeneous data. Moreover, it is yet to be established whether LLMs are capable of precise numerical reasoning, *e.g.* forecasting precise coordinate values that are demanded by motion planning.

To this end, we propose a novel approach that successfully unleashes the power of LLMs to address the motion planning problem in autonomous driving. The critical insight is that we can reformulate motion planning as a language modeling problem. Specifically, we propose to tackle the heterogeneous planner inputs by transforming them into unified language tokens, and we instruct a GPT-3.5 model to understand these tokens and then articulate the waypoint coordinates of a future driving trajectory through natural language description. We further elucidate the essence of language modeling in motion planning from the perspective of tokenizers. Moreover, to stimulate the numerical reasoning potential of GPT-3.5, we propose a prompting-reasoning-finetuning strategy, where GPT-3.5 is initially prompted in the context of autonomous driving, and then performs chain-of-thought reasoning to generate sensible outputs, and finally the model is fine-tuned with human driving trajectories to ensure alignments with human driving behaviors. With this strategy, GPT-3.5 is able to forecast highly precise waypoint coordinates with only a centimeter-level error. The chain-of-thought reasoning further enhances transparency in decision-making and makes our approach more interpretable than other learning-based methods. Benefiting from the state-of-the-art GPT-3.5 model, our approach also exhibits good generalization and common sense reasoning ability.

We summarize our contributions as follows:

· We propose GPT-Driver, a GPT-based motion planner, innovatively transforming the motion planning task into a language modeling problem. We also provide an intuitive interpretation of language modeling in motion planning through the lens of the GPT tokenizer.

· We propose a novel prompting-reasoning-finetuning strategy in the context of autonomous driving, which enables precise numerical reasoning and transparent decision-making of our approach.

· Our GPT-Driver demonstrates superior motion planning performance, few-shot generalization ability, and interpretability compared to the state-of-the-art motion planners on the nuScenes dataset.

## 2 RELATED WORKS

**Motion planning in autonomous driving.** Motion planning aims to forecast safe and comfortable driving routes for autonomous vehicles. Existing approaches can be divided into three categories: rule-based, optimization-based, and learning-based methods. The rule-based approaches (Treiber et al., 2000; Thrun et al., 2006; Bacha et al., 2008; Leonard et al., 2008; Urmson et al., 2008; Chen et al., 2015; Sauer et al., 2018; Fan et al., 2018; Dauner et al., 2023) resort to pre-defined rules to determine future driving trajectories. Intelligent Driver Model (Treiber et al., 2000) (IDM) is a seminal work that proposed a heuristic motion model to follow a leading vehicle in traffic while maintaining a safe distance. Despite being simple and interpretable, IDM lacks sufficient capability to handle complicated driving behaviors such as U-turns. The optimization-based approaches (Li et al., 2022; Liniger et al., 2015; Scheffe et al., 2022) formulate motion planning as an optimal control problem. In contrast, the learning-based approaches (Bojarski et al., 2016; Codevilla et al., 2018; 2019; Rhinehart et al., 2019; Zeng et al., 2019; Sadat et al., 2020; Casas et al., 2021; Hu et al., 2022; 2023) proposed to handle complex driving scenarios by learning from large-scale human driving data. Neural motion planner (Zeng et al., 2019) suggested using a learned cost volume to assess each feasible driving trajectory. P3 (Sadat et al., 2020), MP3 (Casas et al., 2021), ST-P3 (Hu et al., 2022), and UniAD (Hu et al., 2023) proposed end-to-end learning of planning and other tasks in autonomous driving. These approaches rely on deep neural networks to predict future driving trajectories, while the decision-making process is implicitly encoded in neural networks and thus less interpretable.

Our GPT-Driver is a learning-based motion planner. In contrast to other learning-based approaches, we leverage the generalization and reasoning ability of the GPT-3.5 model, which enables our model to tackle those long-tailed driving scenarios that are generally challenging to other methods. Our method also has better interpretability thanks to the novel prompting-reasoning-finetuning strategy.

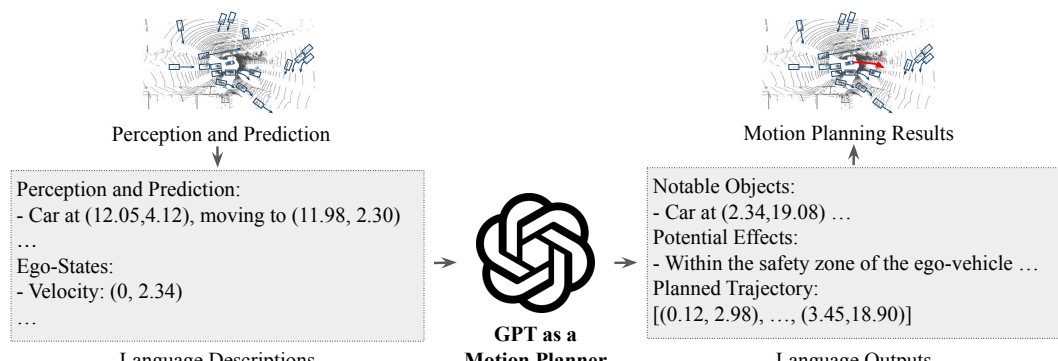

Figure 1: **Overview of GPT-Driver.** We reformulate motion planning as a language modeling problem. We convert observations and ego-states into language prompts, guiding the LLM to produce a planned trajectory alongside its decision-making process in natural language. Subsequently, this planned trajectory is reverted to the numerical format for motion planning.

**Large language models.** Large Language Models (LLMs) are artificial intelligence systems trained on Internet-scale data to understand and generate human-like text, showcasing remarkable abilities in natural language processing. GPT (Brown et al., 2020) is a pioneering work that proposed the Generative Pre-trained Transformer to tackle language understanding and generation problems. The following versions GPT-3.5 and GPT-4 (OpenAI, 2023) demonstrated impressive chatting and reasoning ability. LLaMA and LLaMA 2 (Touvron et al., 2023a;b) are open-source foundation language models. To better harness the capabilities of LLMs, InstructGPT (Ouyang et al., 2022) proposed to train LLMs to follow instructions with human feedback. (Wei et al., 2022) proposed chain-of-thought prompting to enhance the reasoning ability of LLMs. ReAct (Yao et al., 2022) exploited the synergy of reasoning and acting in LLMs. These methods have bolstered the language understanding and decision-making capabilities of LLMs. Despite the success of LLMs in language understanding, exploiting the power of LLMs in autonomous driving remains an open challenge, as the inputs and outputs of autonomous systems are not language. In this paper, we tackle this challenge by reformulating the traditional driving problem into a language modeling problem. Moreover, we propose a novel prompting-reasoning-finetuning strategy tailored for autonomous driving, which is significantly different from the existing works (Yao et al., 2022; Wei et al., 2022) and amplifies the reasoning capabilities of the LLM-based planner.

There is also a series of works (Ahn et al., 2022; Fu et al., 2023; Huang et al., 2022; Song et al., 2022) using LLMs for task-level planning, *i.e.*, planning high-level actions for embodied agents. In contrast, our method focuses on motion planning, *i.e.* planning waypoint-based low-level driving trajectories for autonomous vehicles. Unlike the natural language descriptions used for high-level actions, trajectories are represented as sets of numerical coordinates, posing a greater challenge for LLMs. To the best of our knowledge, our work is the first to demonstrate GPT-3.5's capability for detailed numerical reasoning in motion planning.

## 3 GPT-DRIVER

In this section, we present GPT-Driver, an LLM-based motion planner for autonomous driving. An overview of our GPT-Driver is shown in Figure 1. We first introduce the basic concept and problem definition of motion planning in the context of autonomous driving (Section 3.1). Then, we demonstrate how to reformulate motion planning as a language modeling problem (Section 3.2). Finally, we introduce how to address this language modeling problem using a novel prompting-reasoning-finetuning strategy (Section 3.3).

### 3.1 PROBLEM DEFINITION

The objective of motion planning in autonomous driving is to plan a safe and comfortable driving trajectory $\mathcal{T}$ with observations $\mathcal{O}$ and ego-states $\mathcal{S}$ as input. The motion planning process $F$ can be formulated as:

$$\mathcal{T} = F(\mathcal{O}, \mathcal{S}). \tag{1}$$

A planned trajectory $\mathcal{T}$ can be represented as a set of waypoints of $t$ timesteps: $\mathcal{T} \in R^{t \times 2}$:

$$\mathcal{T} = \{(x_1, y_1), \cdots, (x_t, y_t)\}, \tag{2}$$

where $(x_i, y_i)$ is a 2D waypoint coordinate that denotes the vehicle's anticipated location at the timestep $i$. The ego-states $S$ generally consist of a historical trajectory of this vehicle and its current status such as velocity and acceleration. The observations $\mathcal{O}$ contain the outputs of perception and prediction systems, *e.g.*, detected object bounding boxes and their future motions.

The learning-based motion planners generally learn the trajectory $\mathcal{T}$ by imitating a human driver's driving trajectory $\hat{\mathcal{T}}$ with $L1$ regression, where the loss function $\mathcal{L}_{reg}$ can be formulated as:

$$\mathcal{L}_{reg} = \sum_{i=1}^{T} (|x_i - \hat{x}_i| + |y_i - \hat{y}_i|), \tag{3}$$

where $(x_i, y_i)$ and $(\hat{x}_i, \hat{y}_i)$ are waypoints of the planned trajectory $\mathcal{T}$ and the human trajectory $\mathcal{T}'$ respectively. Albeit simple, these approaches attempt to simultaneously regress waypoints across different scales, *e.g.* coordinate values ranging from 0 to over 50, which generally results in imprecise coordinate estimations of the more distant waypoints. To this end, we propose a novel approach that supplants the traditional $L1$ trajectory regression with a language modeling framework.

### 3.2 MOTION PLANNING AS LANGUAGE MODELING

The crucial insight of this paper is to transform motion planning into a language modeling problem. Given a driving trajectory $\mathcal{T}$, we can represent it as a sequence of words that describe this trajectory:

$$\mathcal{T} = K(\{(x_1, y_1), \cdots, (x_t, y_t)\}) = \{w_1, \cdots, w_n\}, \tag{4}$$

where $w_i$ is the $i$-th word in this sequence. Please note that each coordinate value $x$ or $y$ in Equation 2 can be freely transformed into a set of words $\{w\}$ using a language tokenizer $K$. For instance, a coordinate value 23.17 can be transformed into three words: "23", ".", and "17" using the GPT-3.5 tokenizer. With this language representation, we can then reformulate the motion planning problem as a language modeling problem:

$$\mathcal{L}_{LM} = -\sum_{i=1}^{N} \log P(\hat{w}_i | w_1, \cdots, w_{i-1}), \tag{5}$$

where $w$ and $\hat{w}$ are the words from the planned trajectory $\mathcal{T}$ and the human driving trajectory $\hat{\mathcal{T}}$ respectively. By learning to maximize the occurrence probability $P$ of the words $\hat{w}$ derived from the human driving trajectory $\hat{\mathcal{T}}$, motion planners can generate human-like driving trajectories.

We can derive a natural interpretation of how language modeling works in motion planning through the lens of tokenization. Take the coordinate value 23.17 as an example. Through tokenization, it is decomposed into "23" which is the integer part of this value, ".", and "17" which is the decimal part of this value. Hence, the process of predicting this waypoint coordinate is essentially first estimating a coarse location at the meter level ("23" here) and then estimating a fine-grained location at the centimeter level ("17" here). Moreover, the estimations are established by classifications of the correct tokens in the vocabulary, rather than regression of their absolute values.

**Autonomous Driving Planner**
Role: You are the brain of an autonomous vehicle. Plan a safe 3-second driving trajectory. Avoid collisions with other objects.

Context
- Coordinates: X-axis is perpendicular, and Y-axis is parallel to the direction you're facing. You're at point (0,0).
- Objective: Create a 3-second route using 6 waypoints, one every 0.5 seconds.

Inputs
1. Perception & Prediction: Info about surrounding objects and their predicted movements.
2. Ego-States: Your current state including velocity, heading angular velocity, can bus data, heading speed, and steering signal.
3. Historical Trajectory: Your past 2-second route, given by 4 waypoints.
4. Mission Goal: High-level goal for the next 3 seconds.

Task
- Thought Process: Note down critical objects and potential effects from your perceptions and predictions.
- Action Plan: Detail your meta-actions based on your analysis.
- Trajectory Planning: Develop a safe and feasible 3-second route using 6 new waypoints.

Output
- Thoughts:
  - Notable Objects
    Potential Effects
- Meta Action
- Trajectory (MOST IMPORTANT):
  - [(x1,y1), (x2,y2), ... , (x6,y6)]

---

Perception and Prediction:
 - **animal** at (**-1.93**,**7.00**), moving to (**-2.31**,**10.89**).
 - **car** at (**-8.67**,**0.12**), moving to (**-8.50**,**-0.08**).
 - **adult** at (**-1.21**,**6.78**), moving to (**-1.29**,**10.48**).

Ego-States:
 - Velocity (vx,vy): (**0.00**,**1.46**)
 - Heading Angular Velocity (v_yaw): (**-0.00**)
 - Acceleration (ax,ay): (**0.01**,**-0.15**)

Historical Trajectory (last 2 seconds): [(**-0.00**,**-6.74**), (**-0.03**,**-4.73**), (**-0.03**,**-3.07**), (**-0.02**,**-1.46**)]

Mission Goal: **RIGHT**

Figure 2: **An example of input prompts provided to the LLM.** The upper text box offers a universal context related to motion planning for every driving scenario. The lower text box provides a language description of the observations and ego-states specific to this particular frame. Parameterized inputs are highlighted in **red**.

We note that language modeling has been employed in other tasks of computer vision and robotics, such as object detection (Chen et al., 2021; Xue et al., 2022; Wang et al., 2023) and robotic control (Brohan et al., 2023). However, these approaches heavily rely on specially designed tokens and tokenizers, which makes their methods less intuitive and hard to generalize to other tasks. In contrast, our key observation is that a commonly used language tokenizer such as the GPT tokenizer already has sufficient capability to estimate very precise numerical values for motion planning. This unique finding makes our approach significantly simpler than prior methods, and also makes our approach more generalizable and compatible with natural language.

## 3.3 PROMPTING-REASONING-FINETUNING

Despite the potential of language modeling in motion planning, simply adopting (Wei et al., 2022; Ouyang et al., 2022; Yao et al., 2022) and prompting GPT-3.5 to generate trajectories didn't work in practice (See Section 4.5). To this end, we introduce a novel prompting-reasoning-finetuning strategy that stimulates the potential of language modeling to address the motion planning problem. Specifically, we introduce a method that utilizes the GPT tokenizer $K$ to convert observations $\mathcal{O}$ and ego-states $\mathcal{S}$ into language prompts. These prompts are then fed into the GPT-3.5 model $F_{GPT}$. We instruct the model to articulate its decision-making process explicitly and produce planned trajectories $\mathcal{T}$ in natural language. Finally, we fine-tune the GPT model's outputs to ensure alignment with human driving trajectories. The prompting-reasoning-finetuning process can be formulated as

Thoughts:
 - Notable Objects from Perception: **animal at (-1.93,7.00)**
   Potential Effects from Prediction: **within the safety zone of the ego-vehicle at the 1.0-second timestep**
 - Notable Objects from Perception: **adult at (-1.21,6.78)**
   Potential Effects from Prediction: **within the safety zone of the ego-vehicle at the 1.0-second timestep**

Meta Action: **TURN RIGHT WITH A CONSTANT SPEED**

Trajectory:
 **[(0.11,1.14), (0.45,2.28), (1.12,3.47), (2.18,4.54), (3.65,5.29), (5.49,5.58)]**

Figure 3: **An example of the expected outputs of the LLM.** The chain-of-thought reasoning and the planned trajectory are highlighted in **red**.

$$\{\mathcal{T}, \mathcal{R}\} = F_{GPT}(K(\mathcal{O}, \mathcal{S})), \tag{6}$$

where $\mathcal{T} = \{w_1, \cdots, w_n\}$ is a language description of the trajectory in Equation 4, and $\mathcal{R}$ denotes a language description of the chain-of-thought reasoning and decision-making process. In contrast to the traditional motion planning methods that solely generate planned trajectories, our approach generates both the trajectories $\mathcal{T}$ and the explicit reasoning process $\mathcal{R}$, which makes our model's decision-making process more transparent. Hence, our approach demonstrates better interpretability than the existing methods.

In subsequent sections, we delve into details of the prompting, reasoning, and fine-tuning process.

**Prompting.** A key obstacle in using LLMs for motion planning is the disparity in data types: while motion planners process heterogeneous inputs of observations and ego-states, LLMs are primarily designed to handle language inputs. To overcome the above limitations, we resort to the parameterized representations of observations and ego-states and convert them into language descriptions. In particular, we utilize detected objects that are parameterized by their class names and locations as perception results. For each object, we formulate a sentence capturing these attributes. These sentences collectively form the perception prompts. Similarly, we can craft prediction prompts by converting the parameterized future trajectories of detected objects into natural language descriptions. We can also generate the prompts for ego-states by articulating the ego vehicle's current status such as velocity and heading. Furthermore, we provide general context information about motion planning, such as the coordinate system, objective, *etc.* Finally, we rephrase these prompts in a more concise format using ChatGPT-4 and utilize them as the inputs to the GPT-3.5 model. An example of prompts is shown in Figure 2.

**Reasoning.** A common weakness of current motion planners is their limited interpretability, since these planners generate planned trajectories from black-box neural networks without elucidating the reasoning behind their decisions. To address this problem, we propose a novel chain-of-thought reasoning strategy specifically designed for autonomous driving. In particular, we summarize the chain-of-thought reasoning process in autonomous driving into 3 steps: First, from the perception results, the motion planner needs to identify those critical objects that may affect its driving dynamics. Second, by analyzing the future motions of these critical objects from the prediction results, the planner should infer when, where, and how this critical object may influence the ego vehicle. Third, on top of the insights gained from the previous analyses, the planner needs to draw a high-level driving decision and then convert it into a planned trajectory. This three-step reasoning framework offers a more structured approach to motion planning and ensures greater transparency throughout the planning procedure. An example is shown in Figure 3.

**Fine-tuning.** To align the LLM's outputs with human driving behaviors, we employ a simple fine-tuning strategy using the OpenAI fine-tuning API. Specifically, we collect human driving trajectories $\hat{\mathcal{T}}$ for each scenario from driving logs. To generate the ground truth guidance of chain-of-thought reasoning $\hat{\mathcal{R}}$, we initially compute a hypothetical ego-trajectory based on the current velocity and acceleration of the ego vehicle, assuming there is no interference. Then, we identify the critical objects and their potential effects by examining if any objects, based on their present positions and predicted future paths, overlap with the hypothetical ego-trajectory. We found this strategy works well in practice, enabling us to bypass the tedious task of manually annotating the reasoning process. Finally, we can fine-tune the LLM's outputs $\{\mathcal{T}, \mathcal{R}\}$ with the ground truth $\{\hat{\mathcal{T}}, \hat{\mathcal{R}}\}$ using the

language modeling loss $\mathcal{L}_{LM}$ defined in Equation 5. During inference, we transform the language output of a planned trajectory back to its numerical format for evaluation.

## 4 EXPERIMENTS

In this section, we demonstrate the effectiveness, generalization ability, and interpretability of our GPT-Driver through extensive experiments on the large-scale and real-world nuScenes dataset (Caesar et al., 2020). We first introduce the experimental settings and evaluation metrics, and then compare our approach against state-of-the-art motion planning methods on the nuScenes dataset. Finally, we conduct studies to evaluate the generalization and interpretability of our approach.

### 4.1 EXPERIMENTAL SETUP

The nuScenes dataset is a large-scale and real-world autonomous driving dataset. It contains 1000 driving scenarios and approximately 40000 key frames encompassing a diverse range of locations and weather conditions. We follow the general practice in prior works (Hu et al., 2022; 2023; Jiang et al., 2023) and split the whole dataset into training, validation, and testing sets. We use the training set to fine-tune our model and evaluate our model's performance on the validation set, which ensures a fair comparison with prior works.

For a fair comparison with other methods, we adopt the evaluation metrics in UniAD (Hu et al., 2023) to evaluate our planned trajectories. It contains two metrics: L2 error (in meters) and collision rate (in percentage). The average L2 error is computed by measuring each waypoint's distance in the planned and ground-truth trajectories. It reflects the proximity of a planned trajectory to a human driving trajectory. The collision rate is computed by placing an ego-vehicle box on each waypoint of the planned trajectory and then checking for collisions with the ground truth bounding boxes of other objects. It reflects the safety of a planned trajectory. We follow the common practice in previous works and evaluate the motion planning result in the 3-second time horizon.

### 4.2 COMPARISON AGAINST THE STATE-OF-THE-ART METHODS

End-to-end driving approaches like UniAD (Hu et al., 2023) perform motion planning based on their internal perception and prediction outputs. For a fair comparison with this work, we build our model on top of the perception and prediction results from their model. We also tried leveraging the perfect perception and prediction results from the dataset for motion planning.

Table 1 shows the motion planning performance of our GPT-Driver against the state-of-the-art methods. It is clear that our GPT-Driver significantly outperforms the prior works in the L2 metric by a large margin, demonstrating the effectiveness of our approach in generating human-like driving trajectories. L2 is a strong indicator of the imitation learning ability of motion planners. Our approach surpasses the state-of-the-art approaches in L2, indicating that the fine-tuned LLM has a stronger imitation learning ability compared to MLP-based planners. The collision rate serves as a strong indicator of the safety of motion planning. Our approach also aligns closely with the state-of-the-art methods in the collision metric, indicating our capability to plan safe driving trajectories. Please note that other baseline methods heavily rely on tricks such as post-optimization to lower the collision rate. By contrast, our approach doesn't rely on these tricks. Moreover, when replacing the perfect perception and prediction with the learned ones, the planning performance only drops slightly, which indicates the robustness of our GPT-Driver to perception and prediction errors. It is worth noting that these state-of-the-art planners (Hu et al., 2023) heavily rely on dense occupancy grids and maps, in addition to detection and prediction, which makes their systems intricate and time-consuming. In contrast, our approach only takes language descriptions of detections and predictions as input observations, which is much simpler than prior methods. Our method also has the potential to incorporate vectorized maps to further boost the performance.

### 4.3 FEW-SHOT MOTION PLANNING

To further validate the generalization ability of our GPT-Driver, we designed a few-shot motion planning experiment. Specifically, we sampled $1\%$, $10\%$, $50\%$ of the training scenarios and utilized them for fine-tuning our model and training the state-of-the-art motion planner in UniAD. For a fair

| Method | L2 (m) ↓ | | | | Collision (%) ↓ | | | |
|---|---|---|---|---|---|---|---|---|
| | 1s | 2s | 3s | Avg. | 1s | 2s | 3s | Avg. |
| NMP (Zeng et al., 2019) | - | - | 2.31 | - | - | - | 1.92 | - |
| SA-NMP (Zeng et al., 2019) | - | - | 2.05 | - | - | - | 1.59 | - |
| FF (Hu et al., 2021) | 0.55 | 1.20 | 2.54 | 1.43 | 0.06 | 0.17 | 1.07 | 0.43 |
| EO (Khurana et al., 2022) | 0.67 | 1.36 | 2.78 | 1.60 | **0.04** | **0.09** | 0.88 | 0.33 |
| ST-P3 (Hu et al., 2022) | 1.33 | 2.11 | 2.90 | 2.11 | 0.23 | 0.62 | 1.27 | 0.71 |
| UniAD (Hu et al., 2023) | 0.48 | 0.96 | 1.65 | 1.03 | 0.05 | 0.17 | 0.71 | **0.31** |
| GPT-Driver[†] | **0.21** | **0.43** | **0.79** | **0.48** | 0.16 | 0.27 | **0.63** | 0.35 |
| GPT-Driver[‡] | 0.20 | 0.42 | 0.72 | 0.45 | 0.14 | 0.25 | 0.60 | 0.33 |

Table 1: **Motion planning performance compared to the state-of-the-art methods.** †: Using perception and prediction results from UniAD. ‡: Using perfect perception and prediction from dataset annotations. Our approach significantly outperforms prior works by a large margin in L2 and performs on par with the top methods in collision rate.

| Method | Avg. L2 (m) ↓ | | | | Avg. Collision (%) ↓ | | | |
|---|---|---|---|---|---|---|---|---|
| | 1% | 10% | 50% | 100% | 1% | 10% | 50% | 100% |
| UniAD (Hu et al., 2023) | 5.37 | 1.80 | 1.42 | 1.03 | 6.86 | 1.31 | 0.49 | **0.31** |
| GPT-Driver | **0.84** | **0.60** | **0.54** | **0.48** | **0.64** | **0.45** | **0.37** | 0.35 |

Table 2: **Few-shot motion planning results** compared to the state-of-the-art planner UniAD. Our approach performs significantly better than UniAD when the training data is limited and demonstrates better generalization ability.

comparison, both UniAD and our approach leverage the same pretrained detection and prediction modules as inputs, and all other parameters remain the same. Table 2 illustrates the few-shot motion planning results. Our approach attains decent motion planning results on the validation set when exposed to only $10\%$ of the full training scenarios, while UniAD failed to obtain good performance when the training data is limited. In contrast to other learning-based planners that heavily rely on large amounts of data, our GPT-Driver fine-tuned on a few training scenarios could generalize well to the full validation set, which indicates its strong generalization and few-shot learning ability.

## 4.4 INTERPRETABILITY

To demonstrate the interpretability of our GPT-Driver, we visualized the reasoning outputs and the planned trajectories of our model in Figure 4. From the figure, we can observe that our method is able to identify critical objects and assess their potential effects from all perception and prediction inputs, and then based on these observations it can generate a coherent high-level action as well as a sensible driving trajectory. For example, in the first sub-figure, our GPT-Driver could identify all obstacles such as barriers and traffic cones, and further neglect the far-away white bus that has no effect on our driving route. Then it can generate a turn-right action with a deceleration to avoid collisions with these obstacles. Finally, it plans a smooth and safe turning trajectory. In contrast to previous methods that only generate planned trajectories, our approach generates not only the trajectories but also the reasoning process of how it predicts these trajectories. Thus our approach can demonstrate better interpretability.

## 4.5 FINE-TUNING VS. IN-CONTEXT LEARNING

In-context learning and fine-tuning are two prevalent strategies to instruct an LLM for specific tasks. While our fine-tuning strategy works well in motion planning, it raises the question of whether in-context learning could achieve comparable results in this task. To answer this question, we designed an in-context learning experiment where we used both the inputs and the expected outputs in the

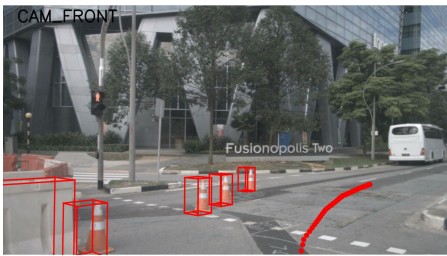

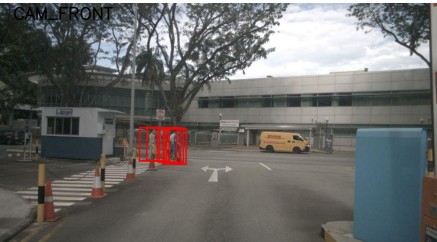

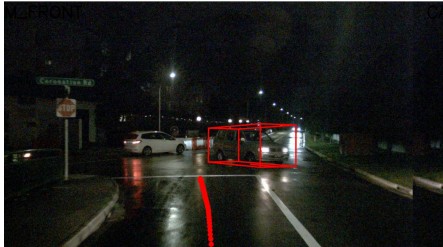

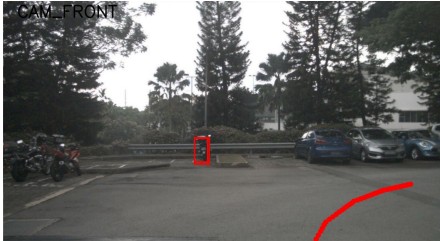

Figure 4: **Visualization** of the GPT-Driver's outputs (text boxes on the right) on the validation set. Planned trajectories and notable objects are highlighted accordingly in **red** on the left images. Please note that the images are only for illustration and are never used in our approach. The visualizations indicate that our method can effectively recognize critical objects and their potential impact from all perception and prediction inputs, and subsequently plan a sensible driving trajectory.

| Method | L2 (m) ↓ | | | | Collision (%) ↓ | | | |
|---|---|---|---|---|---|---|---|---|
| | 1s | 2s | 3s | Avg. | 1s | 2s | 3s | Avg. |
| GPT-Driver (in-context learning) | 2.41 | 3.11 | 4.00 | 3.17 | 4.20 | 5.13 | 6.58 | 5.30 |
| GPT-Driver (fine-tuning) | **0.21** | **0.43** | **0.79** | **0.48** | **0.16** | **0.27** | **0.63** | **0.35** |

Table 3: **Design choices of in-context learning and fine-tuning**. The results indicate fine-tuning is a more effective strategy for instructing the LLM in motion planning.

training set as new exemplar inputs to instruct the LLM. The results in Table 3 suggest that fine-tuning performs significantly better than in-context learning. This is mainly because the model's context window is quite limited in in-context learning, *e.g.* GPT-3.5 can accommodate a maximum of only 5 exemplar inputs every time in our case. Hence, our fine-tuning strategy is indispensable.

### 4.6 LIMITATIONS

Due to the limitations of the OpenAI APIs, we are unable to obtain the inference time of our model. Thus it remains uncertain whether our approach can meet the real-time demands of commercial driving applications. Typically, the GPT-based planner would exhibit a longer inference time compared to existing MLP-based planners. Nevertheless, we argue that there are many techniques that could resolve this problem, *e.g.* distilling a smaller LLM, *etc.* We leave this for future work.

Another limitation lies in the evaluation of motion planning. As open-loop motion planning doesn't fully emulate error accumulation in the driving process, recently close-loop motion planning has become increasingly popular to evaluate the performances of motion planners. We leave close-loop motion planning of our GPT-Driver for future work.

## 5 CONCLUSION

In this paper, we introduce GPT-Driver, an innovative method that transforms the OpenAI GPT-3.5 model into a dependable motion planner for autonomous driving. We reformulate motion planning as a language modeling problem, and we propose a novel prompting-reasoning-finetuning strategy to tackle this problem. Through extensive experiments on the large-scale autonomous driving dataset, our approach has demonstrated superior planning performance, generalization, and interpretability compared to existing works. Future works include optimizing the inference time and involving more sensor observations such as high-definition maps in input prompts.

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
