# OpenReview forum: "GPT-Driver: Learning to Drive with GPT"
_ICLR.cc/2024/Conference — Submitted to ICLR 2024_

### Official Review · Reviewer_PY8W · 2023-10-30

**Soundness:** 3 good
**Presentation:** 3 good
**Contribution:** 3 good
**Rating:** 5
**Confidence:** 5

**Summary:**

The author proposed a scheme utilizing ChatGPT for motion planning, where perception results (or ground truth) are used as inputs. With meticulous prompt design and finetuning through the API, the approach achieved commendable open-loop performance on the large-scale Nuscenes dataset. Additionally, experiments were designed to demonstrate the large language model's generalizability (few-shot) and interpretability (reasoning) for motion planning tasks.

**Strengths:**

1. The overall text flows smoothly and is easy to understand.
2. Seeing the large language model demonstrate impressive performance and few-shot capabilities in such a simple manner is astonishing. This can better motivate people to continue validating the exploration of large language models in the direction of motion planning.
3. The author will open-source the code, and since the experiment is relatively simple, I believe there is a high probability that the experiment can be reproduced. The confidence level in the experiment's results is very high.

**Weaknesses:**

1. While I personally appreciate the simplicity and effectiveness of this research, I do not believe it possesses sufficient novelty to be a paper for ICLR. The article simply utilizes the ChatGPT API for finetuning, which can be regarded as an application experiment report on motion planning using ChatGPT. Since it does not introduce any new modules or methodologies, I think its actual contribution to the field is quite limited.
2. Motion planning ultimately needs to be validated in a closed-loop system, as the results from open-loop and closed-loop scenarios are not always aligned, as highlighted in [1]. In cases where perception results (or ground truth) are directly used as input, it is easy to integrate into a closed-loop system, and there are numerous readily available public datasets for closed-loop testing, such as Nuplan, Waymo Motion Dataset, MetaDrive, etc. If closed-loop results could be obtained, I believe it would significantly enhance the credibility and contribution of the paper.
3. The descriptions in the RELATED WORKS section of the article are somewhat inaccurate. Dauner et al. (2023)[1] is actually a rule-based method.

[1] Dauner, D., Hallgarten, M., Geiger, A., & Chitta, K. (2023). Parting with Misconceptions about Learning-based Vehicle Motion Planning. arXiv preprint arXiv:2306.07962.

**Questions:**

1. I hope to see closed-loop results; please refer to the "Weaknesses" section for more details.

---

> ### Author Response · Authors · 2023-11-22
> **Thanks for the reviewer's feedback**
>
> Thanks for the reviewer's feedback!
>
> **While I personally appreciate the simplicity and effectiveness of this research, I do not believe it possesses sufficient novelty to be a paper for ICLR. The article simply utilizes the ChatGPT API for finetuning, which can be regarded as an application experiment report on motion planning using ChatGPT. Since it does not introduce any new modules or methodologies, I think its actual contribution to the field is quite limited.**
>
> We believe our method has sufficient novelty to be a paper for ICLR for the following reasons:
>
> (1) Our proposed motion planning as language modeling is a fundamental change to the current regression-based planning methods. As we mentioned in Section 3.2, motion planning as language modeling is **essentially transforming the typical regression problem into a coarse-to-fine region classification problem**, which overcomes the weaknesses of regression (e.g. sensitive to scale variation). Hence we argue that we did introduce new methodologies in our paper.
>
> (2) From the architecture's perspective, our approach **transforms the conventional MLP-based planner by introducing fine-tuned LLM as a motion planner**, which is a fundamental architectural change. Compared to the MLP-based planner, our approach can benefit from the LLM and demonstrates superior generalization ability (see the few-shot learning results) and interpretability, thus enabling a more robust and transparent decision-making process.
>
> (3) Our observation that **LLMs can perform low-level motion planning and estimate very accurate numerical coordinates** is also unique. While a lot of papers are working on leveraging LLMs for language-based task planning, we are the first to demonstrate that LLMs can also perform motion planning and accurate numerical predictions. We believe this observation also brings new insights and merits to the research community.
>
> Overall, the proposed method is simple and can be implemented in a few lines of code. Although the idea is intuitive, the learning objective, architecture, and observations are unique and completely different from existing approaches. Hence, we believe our method has sufficient technical contributions to be accepted to ICLR.
>
> **Motion planning ultimately needs to be validated in a closed-loop system, as the results from open-loop and closed-loop scenarios are not always aligned, as highlighted in [1]. In cases where perception results (or ground truth) are directly used as input, it is easy to integrate into a closed-loop system, and there are numerous readily available public datasets for closed-loop testing, such as Nuplan, Waymo Motion Dataset, MetaDrive, etc. If closed-loop results could be obtained, I believe it would significantly enhance the credibility and contribution of the paper.**
>
> We appreciate the review's suggestion and we also agree with this point. We're now working on adapting GPT-Driver to close-loop evaluation on nuPlan. Since nuPlan is much larger than nuScenes, we are unable to get the close-loop performance in this short discussion phase. We will update the paper once we have finished the experiments.
>
> **The descriptions in the RELATED WORKS section of the article are somewhat inaccurate. Dauner et al. (2023)[1] is actually a rule-based method.**
>
> From our understanding, Dauner et al. (2023)[1] is a hybrid approach that leverages rule-based IDM to generate an initial trajectory and then applies a learning-based neural network to refine the trajectory. For better clarification, we have moved this literature to the rule-based method according to the review's feedback.

---

> ### Comment · Reviewer_PY8W · 2023-11-22
> **Thanks for your response**
>
> Thank you for your comprehensive response to my review. I appreciate the effort you've made to address the concerns raised.
>
>
> While I acknowledge the novelty and simplicity of your approach in using ChatGPT for motion planning, I feel that the depth of the research is somewhat limited. The primary method seems to hinge on integrating ChatGPT API for in-context learning and fine-tuning. Although this is an innovative use of ChatGPT, I believe that for a submission to ICLR, more in-depth analysis and exploration are required to truly advance the field.
>
> Here are some directions that might add depth to your work:
>
> * Algorithmic Innovations: Beyond using the API, consider developing algorithmic improvements or adaptations specific to motion planning.
> * In-Depth Analysis: Explore the limitations and strengths of using LLMs in this context in more detail. How does the LLM handle edge cases or particularly complex driving scenarios? (i.e errors from upstream perception module or complex scenarios.)
> * Comparative Studies: It might be beneficial to compare the performance of your approach against traditional motion planning algorithms under various conditions.
>
>
> Additionally, I strongly agree that results from closed-loop scenarios, especially on platforms like nuPlan, would be more indicative of the practical utility of your method. I am looking forward to seeing the closed-loop performance scores on nuPlan, as these would significantly enhance the credibility and real-world applicability of your research.
>
>
> Once again, thank you for considering my feedback, and I am eager to see how this work evolves with these additional aspects.

---

> ### Author Response · Authors · 2023-11-22
> **Thanks for your advice**
>
> Thanks for your feedback. We sincerely appreciate your advice and will incorporate a more in-depth analysis into our paper.
>
> We still want to emphasize again the impact of our paper:
>
> **Algorithmic Innovation:** It is not merely an innovative use of ChatGPT, but also sheds light on the potential of LLMs to perform accurate numerical reasoning, which has broader impacts on not only motion planning in autonomous driving but also planning and controls for general robotics. We believe the findings in this paper are very significant in autonomous driving and robotics, in the LLM era.
>
> **In-Depth Analysis:** We provided few-shot learning ablation study in the main text and that is another evidence that shows LLMs exhibits strong generalization to little training data. In addition, we have a comparison of using UniAD perception and/or ground-truth perception, and results (Table 1) indicate that our method is robust under imperfect perception outputs.
>
> **Comparative Studies:** We compare to many learning-based motion planning baselines (see Table 1). Due to lack of implementations of traditional motion planning methods, we can't compare to them. Due to time limit and an intensive CVPR submission, we didn't have enough time to re-implement traditional motion planning algorithms.
>
> Once again, we thank the reviewer for the fruitful discussions. We are happy to address any concerns you still have during the discussion period. If the responses are sufficient, we kindly ask that you consider raising your score.

---

### Official Review · Reviewer_1ZwR · 2023-10-31

**Soundness:** 2 fair
**Presentation:** 1 poor
**Contribution:** 2 fair
**Rating:** 5
**Confidence:** 3

**Summary:**

The authors propose some techniques for using ChatGPT3.5 to generate driving trajectories (as a list of coordinates) from a language representation of the world/ego state. The approach outperforms end-to-end learning-based approaches on a computer vision driving benchmark.

**Strengths:**

- How to best utilize LLMs for autonomous vehicles is an interesting and timely question that I think is of great interest to the community.
- That the authors achieved such a huge improvement with fine tuning is a surprising insight (going from worst to best results), but potentially useful if it checks out.

**Weaknesses:**

- The benchmark comparison seems a bit apples to oranges. You compare against end-to-end computer vision approaches from CVPR, but your approach assumes the object detections are given and only solves the planning part. Even if you use the detections from a competing CV method, it is unclear to me how strong this result is as planning is perhaps not the main focus of their approach. It would have been good to have some conventional planning stack as baseline as well.
- Your portrayal of related work outside of CV/learning seems weak/dated with the most recently cited planning paper being from 2018. I quite frequently see conventional planning/optimization papers for autonomous driving at other venues (robotics, AI...). Check some recent surveys, I include one which does not use your terminology of "rule-based" for these either (not sure what you mean by that). [1] S. Teng et al., "Motion Planning for Autonomous Driving: The State of the Art and Future Perspectives," in IEEE Transactions on Intelligent Vehicles, vol. 8, no. 6, pp. 3692-3711, June 2023, doi: 10.1109/TIV.2023.3274536.
- This really is "GPT"-driver, it just uses the web APIs for ChatGPT for prompting and fine tuning. GPT is state of the art (at least if you had used 4.0) so is somewhat defensible, but it would have been interesting to see how this generalized across other LLMs.

Minor questionable claims or presentation issues:

- You write "Albeit simple, these approaches attempt to simultaneously regress waypoints across different scales, e.g. coordinate values ranging from 0 to over 50, which generally results in imprecise coordinate estimations of the more distant waypoints": This seems like it would be fixed by a simple rescaling, is this really a fundamental problem with IL approaches? This also does not mention RL-based approaches (see [1])
- Sec 3.2:  Does all of the IL approaches really use absolute value loss, that seems oddly specific? The discussion of how a number is encoded as a text string seems pretty obvious/trivial.
- "It is worth noting that these state-of-the-art planners heavily rely on multiple heterogeneous observations such as detections, predictions, occupancy grids, and maps, which makes their systems intricate and time consuming." You rely on their detections (maybe predictions, unclear) so this seems at least half misleading.

**Questions:**

- What data did you use for fine tuning vs evaluation? I am not very familiar with this particular benchmark, but your trajectory prediction errors are surprisingly low. The optimal trajectory for driving is in reality sometimes ambigious, possibly multi-modal and an open research problem, so it seems a bit suspicious that you can get centimeter precision on trajectory prediction.
- Can you clarify what prompts you used in your three-stage approach for training. Fig 3. seems to only include the prompt for the final stage?
- Isn't your information about the obstacles both incomplete and technically cheating (acausal) in the simple prompting baseline, since you do not include obstacle velocities, but seemingly where they will be in the future? Care to comment on what this description really means, where they will stop / be x seconds? in the future?

---

> ### Author Response · Authors · 2023-11-21
> **Thanks for the reviewer's feedback**
>
> Thanks for the reviewer's feedback!
>
> **The benchmark comparison seems a bit apples to oranges. You compare against end-to-end computer vision approaches from CVPR, but your approach assumes the object detections are given and only solves the planning part. Even if you use the detections from a competing CV method, it is unclear to me how strong this result is as planning is perhaps not the main focus of their approach. It would have been good to have some conventional planning stack as baseline as well.**
>
> The benchmark comparison is **clearly not apples to oranges** for the following reasons:
>
> (1) We are comparing against end-to-end driving approaches in CVPR mainly because these papers are the state-of-art approaches on the nuScenes motion planning benchmark.
>
> (2) Our approach assumes the object detections are given because our approach focuses on motion planning. As we mentioned in the experiments, we also leverage the same detection outputs as UniAD. Hence from the perspective of motion planners, our approach and UniAD have the same detection inputs, which is a fair comparison.
>
> (3) Planning is the main focus and the ultimate optimization goal of these end-to-end driving methods. The title of UniAD is "Planning-oriented Autonomous Driving", and all the detection and prediction modules are optimized for better planning. Hence, it is clear that motion planners in these baseline methods are strong enough for comparison. Our comparison is completely fair with these baselines.
>
> (4) We agree with the point that it's good to have conventional planning baselines. We didn't compare with conventional planning mainly because there are no published convention planning methods on the nuScenes motion planning benchmark. We will try reproducing some approaches on the nuScenes benchmark in the future.
>
> **Your portrayal of related work outside of CV/learning seems weak/dated with the most recently cited planning paper being from 2018. I quite frequently see conventional planning/optimization papers for autonomous driving at other venues (robotics, AI...). Check some recent surveys, I include one which does not use your terminology of "rule-based" for these either (not sure what you mean by that). [1] S. Teng et al., "Motion Planning for Autonomous Driving: The State of the Art and Future Perspectives," in IEEE Transactions on Intelligent Vehicles, vol. 8, no. 6, pp. 3692-3711, June 2023, doi: 10.1109/TIV.2023.3274536.**
>
> Learning-based and rule-based planning are two mainstream motion planning approaches in the context of autonomous driving. In the survey paper "Motion Planning for Autonomous Driving: The State of the Art and Future Perspectives" you have provided, they focus on introducing planning methods based on imitation learning and reinforcement learning, which has already been covered in this paper. For other optimization-based planning approaches, we believe you're referring to control methods, which aim to output more low-level control signals to trace the trajectory and avoid collisions. Optimization-based control is actually a downstream task to motion planning and is out of the scope of this paper.
>
> **This really is "GPT"-driver, it just uses the web APIs for ChatGPT for prompting and fine tuning. GPT is state of the art (at least if you had used 4.0) so is somewhat defensible, but it would have been interesting to see how this generalized across other LLMs.**
>
> We agree with this point. We are working on fine-tuning llama 2 for motion planning, and we will update the results soon.
>
> **You write "Albeit simple, these approaches attempt to simultaneously regress waypoints across different scales, e.g. coordinate values ranging from 0 to over 50, which generally results in imprecise coordinate estimations of the more distant waypoints": This seems like it would be fixed by a simple rescaling, is this really a fundamental problem with IL approaches? This also does not mention RL-based approaches (see [1])**
>
> Rescaling mitigates the effects but the errors can be amplified when you rescale the regression outputs back to their real values. Hence it is an innate weakness of leveraging regression to resolve the imitation learning problem.
>
> **Sec 3.2: Does all of the IL approaches really use absolute value loss, that seems oddly specific? The discussion of how a number is encoded as a text string seems pretty obvious/trivial.**
>
> State-of-the-art methods (e.g., UniAD, ST-P3) on nuScenes rely on the regression of real-world coordinates. Our discussion reveals the essential difference between regression and language modeling in imitation learning for motion planning, and our language modeling approach is simple to implement and demonstrates promising results compared to those regression-based methods, which we believe is a major contribution and not trivial.

---

> ### Author Response · Authors · 2023-11-21
> **Cont'd**
>
> **"It is worth noting that these state-of-the-art planners heavily rely on multiple heterogeneous observations such as detections, predictions, occupancy grids, and maps, which makes their systems intricate and time consuming." You rely on their detections (maybe predictions, unclear) so this seems at least half misleading.**
>
> These state-of-the-art planners, such as UniAD, take compact maps (H xW grids) and occupancy (H x W x T grids), in addition to perception and prediction, as inputs to their motion planner. Extracting those compact maps and occupancy grids makes their system intricate and time-consuming. By contrast, our approach only leverages text-based perception and prediction (instead of compact maps and occupancy grids) as inputs to GPT-Driver, which is more lightweight compared to these methods. We have revised this part for better clarification.
>
> **What data did you use for fine-tuning vs evaluation? I am not very familiar with this particular benchmark, but your trajectory prediction errors are surprisingly low. The optimal trajectory for driving is in reality sometimes ambiguous, possibly multi-modal and an open research problem, so it seems a bit suspicious that you can get centimeter precision on trajectory prediction.**
>
> As we mentioned in Section 4.1, we leverage the training set of nuScenes for fine-tuning our GPT-Driver, and the validation set for evaluating our approach. This is a commonly adopted training and evaluation protocol on nuScenes and ensures a fair comparison with the baselines on the nuScenes dataset. Trajectory prediction errors are low compared to the baselines, demonstrating the effectiveness of our approach. Although trajectory prediction can be multi-modal, on the nuScenes benchmark we focus on the most likely mode/trajectory and computing the errors w.r.t human driving trajectory.
>
> The results are fully validated on widely used benchmarks and we have followed the same training and evaluation protocol to ensure a fair comparison with baselines. We argue that the reviewer's current suspicion is not based on facts and thus not convincing.
>
> **Can you clarify what prompts you used in your three-stage approach for training. Fig 3. seems to only include the prompt for the final stage?**
>
> We only fine-tune the GPT-Driver **once**. The prompting-reasoning-finetuning strategy means that we design proper prompts as inputs to the GPT-Driver, we enumerate the reasoning process as one of the learning targets, and we fine-tune the LLM using the prompts as inputs and the reasoning target as one of the outputs. Hence, Figure 2 and Figure 3 already include all the inputs and outputs to GPT-Driver during fine-tuning.
>
> **Isn't your information about the obstacles both incomplete and technically cheating (acausal) in the simple prompting baseline, since you do not include obstacle velocities, but seemingly where they will be in the future? Care to comment on what this description really means, where they will stop / be x seconds? in the future?**
>
> Clearly, the information about obstacles **is not technically cheating**. As we mentioned, we transform **both detection and prediction** results into text descriptions. This means we describe not only the current location of the object but also the future moving trajectory (in 6 waypoint coordinates) of this object. In this paper, we simplify the prediction by describing the location where the object ends up in the 3rd second. Prediction, or motion forecasting, is a commonly adopted sub-task in the context of autonomous driving, Most state-of-the-art driving systems rely on a perception-prediction-planning paradigm where detection and prediction results are leveraged as inputs to the motion planners. Here we adopted the detection and prediction results from the state-of-the-art UniAD to ensure a fair comparison with this baseline approach.

---

> > ### Comment · Reviewer_1ZwR · 2023-11-22
> > **Response to authors**
> >
> > I thank the authors for the clarifications which resolved some concerns.
> >
> > However, some concerns remain:
> >
> > **About the credibility of the results:**
> > I still think it is a fair point to ask why you are proposing a pure planning approach but only evaluating it on a benchmark for end-to-end approaches, without comparing to any conventional planning approach. Being SOTA on nuScenes is not necessarily the same thing as being the best planning approach. While actual SOTA in AD is a bit murky due to private sector R&D, the major players at least started from a conventional, modular autonomy stack and I suspect there is still a lot of that in there. The public comment about the recent issues discovered with nuScenes benchmarking reinforces my concern that this is not a mature benchmark for planning, as is the question of the actual L2 metric used. I will await replies on these points before making my final decision.
> >
> > **Minor: About incomplete/acausal ("cheating") inputs:**
> > Thank you for the clarification that the "moving to (x, y)" input actually is from a prediction module and not GT. I still think this is incomplete data as you never seem to tell the planner exactly how many seconds into the future the x and y in "moving to (x,y) represents (i.e. you cannot recover the object velocity without that). However, this can only make your results worse so it's not a big issue for the review.
> >
> > Your prompt just says:
> > "Inputs
> > 1. Perception & Prediction: Info about surrounding objects and their predicted movements."
> >
> > **Minor: Regarding AD related work and terminology:** (minor)
> > You seem to be right about the "rule-based" being accepted terminology for conventional motion planning approaches in the AD community, my bad. I still think the related work section for these approaches is rather weak as it ends in 2018 but as you are focusing on end-to-end approaches it is perhaps good enough. For future reference, even though the survey I posted focused more on learning-based approaches, they do provide a decent overview of local planning in Sec II. Optimization-based does not just mean control either, to consider vehicle kinematics and dynamics in a motion planning problem you often formulate it as an OCP and approximate it with e.g. lattice (graph) or tree planners using motion primitives.
> >
> > Before adjusting my score I will await the outcome of the other discussions, and in particular your response to the issues raised in the public comment.

---

> ### Author Response · Authors · 2023-11-22
> **Thanks for your feedback**
>
> **I still think it is a fair point to ask why you are proposing a pure planning approach but only evaluating it on a benchmark for end-to-end approaches, without comparing it to any conventional planning approach.**
>
> nuScenes is a commonly used **motion planning** benchmark where numerous works have been using this benchmark to evaluate the planning quality of their approach. Again, we'd like to emphasize that we are comparing with the motion planners in these end-to-end driving approaches, instead of directly comparing with end-to-end approaches. We don't compare with the conventional planning approaches because there is no such method implemented or evaluated on nuScenes, but we will try to reproduce some conventional approaches according to your advice.
>
> **The major players at least started from a conventional, modular autonomy stack and I suspect there is still a lot of that in there**
>
> As far as we know, nuScenes is one of the most popular benchmarks for modular autonomy stack (detection, prediction, planning), and we are actually starting from this modular autonomy stack (but not conventional because we're focusing on learning-based approaches).
>
> **The public comment about the recent issues discovered with nuScenes benchmarking reinforces my concern that this is not a mature benchmark for planning, as is the question of the actual L2 metric used.**
>
> We will update the results for all metrics very soon. The differences in L2 calculation do not affect the conclusion though. Again, as we mentioned, nuScenes is a mature benchmark for open-loop evaluation of motion planning, which is agreed by most reviewers.
>
> **Minor: About incomplete/acausal ("cheating") inputs:**
>
> First, it is not an acausal ("cheating") input. As we mentioned in the response, we're taking the prediction module's outputs as inputs to the planner, which is a common practice in most existing works. Second, it is not an incomplete input, as we have already included the start location and the end location of an object in a 3-second prediction horizon (see prompts in Figure 2 for context).
>
> **Regarding AD related work and terminology**
>
> We have included the most recent rule-based work (Dauner et al. 2023) in our paper. We have added the optimization-based approaches you mentioned in the paper.
>
> We sincerely thank you for providing feedback to help us make this submission stronger. Please let us know if our responses address your concerns. We are more than happy to address any further questions during the discussion period. If the responses are sufficient, we kindly ask that you consider raising your score. Thank you so much!
>
> Dauner, D., Hallgarten, M., Geiger, A., & Chitta, K. (2023). Parting with Misconceptions about Learning-based Vehicle Motion Planning. arXiv preprint arXiv:2306.07962.

---

### Official Review · Reviewer_FVv6 · 2023-10-31

**Soundness:** 2 fair
**Presentation:** 3 good
**Contribution:** 2 fair
**Rating:** 5
**Confidence:** 3

**Summary:**

This work studied the application of LLM in motion planning for autonomous driving. In the proposed GPT-Driver framework, perception and prediction results (e.g., object types, coordinates, and predicted future coordinates) together with the ego states are converted into language tokens. Then, they are used to prompt an LLM to produce a planned trajectory alongside its decision-making process in natural language. In particular, the authors propose a fine-tuning scheme with auto-generated reasoning labels to fine-tune a GPT-3.5 model for the purpose of motion planning. The results show that the GPT-Driver outperforms existing learning-based motion planners in terms of imitating human drivers and performs on par with top methods in collision rate.

**Strengths:**

1. The proposed method is simple, straightforward, and well-performing in the studied driving scenarios.
2. It provides informative insights on the feasibility and performance of steering LLMs into motion planners producing numerical waypoints. It is particularly interesting and promising that the authors show that GPT-Driver can outperform existing approaches after few-shot fine-tuning.

**Weaknesses:**

While the proposed GPT-Driver has demonstrated impressive performance, the paper lacks in-depth analysis to help the audience gain a deeper understanding of the model's performance and limitations:

1. For example, an ablation study should be conducted to evaluate the benefit of having chain-of-thought reasoning in the LLM's output. While it is well-known that chain-of-thought reasoning boosts LLM's performance, it is worth evaluating its contribution to the motion planning task.

2. Also, there should be an ablation study to evaluate the benefit of having the auto-generated chain-of-thought reasoning labels during fine-tuning. While the proposed method to auto-label the chain-of-thought reasoning through hypothetical ego-trajectory is sensible, it is not guaranteed to generate the ground-truth reasoning process (i.e., identifying the actual causal objects and their relations to the ego agents). The authors claimed that this strategy worked well in practice. I wonder how the authors evaluated the quality of the auto-generated labels and drew such a conclusion. There should be numerical results to examine the quality of the auto-generated labels, and an ablation study to validate that the fine-tuning process indeed benefits from the plausibly noisy and inaccurate reasoning labels.

3. The GPT model is only prompted with a simplified textual description of the traffic scene, e.g., without maps, historical trajectories of the objects, or predicted future trajectories over multiple timesteps. It is quite surprising that the GPT model can surpass carefully designed learning-based planners by a large margin in L2 errors. It is rather counter-intuitive as the information currently missing (e.g., maps, historical contexts, future trajectories) is normally considered important for motion planning in autonomous driving. The authors should provide an in-depth analysis and pinpoint the scenarios where GPT-Driver has clear advantages against SOTA methods and cases where GPT-Driver suffers. Only showing the average L2 errors and collision rates could be misleading, as the average statistics highly depend on the data distribution.

**Questions:**

1. Could the authors clarify how the hypothetical ego trajectory is generated? Whether an object is identified as critical seems to highly depend on the hypothetical ego trajectory. The authors should discuss how they designed the generation algorithm and adjusted the hyperparameters.

2. Is the motion planning performance evaluated with the most likely output sequence? How stable and reliable is the GPT-Driver in generating sensible reasoning processes and trajectories? Is it able to account for multi-modality in driving behavior?

---

> ### Comment · Reviewer_FVv6 · 2023-11-22
> **Awaiting Author's Response**
>
> It is a friendly reminder that the authors have not uploaded their responses to my review comment yet. Note that, due to the time difference, I may not be able to engage in further discussion if the response is uploaded in the last minutes.
>
> In addition to my original review comments, I agreed with the other reviewers and the public comment on the limitation of open-loop evaluation. I agree with the authors that nuScenes can serve as a mature *open-loop* planning benchmark. However, it does not mean that it is sufficient as a comprehensive benchmark for AD planning, due to the limitation of the open-loop evaluation scheme itself. I think the authors should either incorporate closed-loop evaluation on nuPlan as they promised or carefully interpret the current set of open-loop evaluation results and discuss the limitations, without overly exaggerating the model's performance.
>
> Note that, in one of their responses, the authors said, "Since nuPlan is much larger than nuScenes, we are unable to get the close-loop performance in this short discussion phase. We will update the paper once we have finished the experiments." As I recall, the authors are not allowed to revise the paper after Nov. 22nd. If the authors think they are not able to include the closed-loop evaluation results in the discussion period, I would suggest the authors attempt to address these issues in another way. I cannot accurately re-evaluate the quality of the paper only based on the authors' promises.

---

> > ### Author Response · Authors · 2023-11-22
> > **Thanks for your comments**
> >
> > We're still working on the ablation studies you mentioned. It takes some time before we are able to release the response to you. We sincerely apologize for this inconvenience.
> >
> > For the evaluation of motion planning, we'd like to emphasize that the L2 and collision rate metrics are strong indicators of the motion planning quality.
> >
> > As we mentioned in Section 4.1, L2 reflects the proximity of a planned trajectory to a human driving trajectory. Hence it is a strong indicator of the imitation learning ability of motion planners. Our approach surpasses the state-of-the-art approaches in L2, indicating that fine-tuned LLM has stronger imitation learning ability compared to MLP-based planners.
> >
> > The collision rate is more aligned with close-loop results and serves as a strong indicator of the safety of a planned trajectory. Our approach performs on par with the baseline approaches in collision rate, indicating that our approach is able to generate a safe trajectory. Please note that other baseline methods heavily rely on tricks such as post-optimization to lower the collision rate. By contrast, our approach doesn't rely on these tricks.
> >
> > We have included the above interpretations (Section 4.2) and discussed the limitations of open-loop motion planning (Section 4.6) in our paper. We sincerely appreciate your advice. Thank you!

---

> ### Author Response · Authors · 2023-11-23
> **Thanks for your advice**
>
> Thanks for your advice!
>
> **For example, an ablation study should be conducted to evaluate the benefit of having chain-of-thought reasoning in the LLM's output.**
>
> Here is the ablation study of removing chain-of-thoughts reasoning:
>
> Method | L2   |      |      |      | Collision |      |      |      |
> |------|------|------|------|------|-----------|------|------|------|
> GPT-Driver  | 1s   | 2s   | 3s   | Avg. | 1s        | 2s   | 3s   | Avg. |
> w/ cot. reason | 0.21 | 0.43 | 0.79 | 0.48 | 0.16      | 0.27 | 0.63 | 0.35 |
> w/o cot. reason | 0.22 | 0.45 | 0.82 | 0.49 | 0.16      | 0.25 | 0.63 | 0.34 |
>
> Removing chain-of-thoughts reasoning slightly harms the L2 but performs slightly better in the collision. We interpret this insignificant synergy as the LLM does not benefit from a joint language modeling of reasoning and trajectory. Since reasoning will account for most of the output tokens, simultaneously optimizing reasoning and trajectory will cause the LLM to pay less attention to trajectory prediction. Recently we found a cascaded design where the LLM first outputs reasoning and then uses reasoning to output trajectory works better.
>
> **Could the authors clarify how the hypothetical ego trajectory is generated? Whether an object is identified as critical seems to highly depend on the hypothetical ego trajectory. The authors should discuss how they designed the generation algorithm and adjusted the hyperparameters.**
>
> The hypothetical ego trajectory is generated by calculating the future waypoint locations based on the current velocity and acceleration (assuming the vehicle maintains the current speed and drives without interference.) We will identify those objects that fall into a 5x3 m region of any waypoint as critical objects. This auto-generation process is essentially which objects will affect driving when we don't take any special actions.
>
> **Is the motion planning performance evaluated with the most likely output sequence? How stable and reliable is the GPT-Driver in generating sensible reasoning processes and trajectories? Is it able to account for multi-modality in driving behavior?**
>
> The LLM only generates one sequence at a time. The outputs of LLMs are quite stable and very few outputs over the whole validation set contain invalid format. It is able to account for multi-modality when we turn up the temperatures of LLMs and perform multiple runs.

---

### Official Review · Reviewer_cPYB · 2023-11-01

**Soundness:** 2 fair
**Presentation:** 3 good
**Contribution:** 2 fair
**Rating:** 5
**Confidence:** 4

**Summary:**

The authors present a novel method for motion planning in the context of autonomous driving, where they propose to use GPT to both output the motion plan and to explain the reasoning behind it. They fine-tune the GPT model using textual representation of the surrounding context and the output motion plan, and show that the method compares very positively when compared to the existing state-of-the-art.

**Strengths:**

- A very relevant problem being evaluated.
- Interesting and novel approach being proposed.
- Promising experimental results.

**Weaknesses:**

- The method does not seem to be very feasible for online execution.
- The method seems to critically depend on the existing SOTA methods as its integral part, making the overall system quite complex.
- The experimental section can be improved.

**Questions:**

I found the work quite interesting, and the combination of the motion planner with GPT seems like a neat idea (although not that novel at this point). However, the method seems far from being actually applicable in the real world, which the authors don't really explore or address (beyond a very brief explanation in Section 4.6 that seems insufficient and handwavy). Moreover, the explanations of the method can be improved significantly, and the methodology itself seems quite complex and dependent on the existing SOTA methods.
Detailed comments can be found below:
- The authors should fix the format of the references. Instead of "P3 (Sadat et al., 2020)" they use "P3 Sadat et al. (2020)" throughout the work, which is incorrect and adds some confusion in several places.
- Figure 1 is not referenced in the text.
- The method assumes the existing strong method for perception and prediction, which seems like quite a large requirement. The input to GPT assumes detections and their predicted trajectories, which seems to add quite a lot of complexity (both from the training and inference standpoint).
- Related to this, the authors don't really do an ablation study of the perception/prediction module, which would give an indication of how robust is GPT to this part of the methodology.
- In eq (2) the authors say that the input to their model is a map, yet that is not the case as they don't provide the map to the model.
- Later they say that their model can indeed take the map as an input, but given that they represent all inputs as a text it is far from clear how can that be done.
- In Section 4.2 the authors say that other approaches depend on various heterogeneous inputs "which makes their systems intricate and time-consuming", yet the proposed method also depends on the same inputs since it depends on UniAD. So the authors are not being really honest in this case.
- In Section 4.3 it is unclear if the authors use fully trained UniAD for generating input strings for their method, or if they use partially trained UniAD. This should be clarified.
- Some sort of latency analysis should be provided, beyond just a handwavy explanation from Section 4.6. This is important for the practical application of their method and is something that the authors should explore.

---

> ### Author Response · Authors · 2023-11-21
> **Thanks for the reviewer's feedback**
>
> Thanks for the reviewer's feedback!
>
> **The authors should fix the format of the references. Instead of "P3 (Sadat et al., 2020)" they use "P3 Sadat et al. (2020)" throughout the work, which is incorrect and adds some confusion in several places.**
>
> Thanks for pointing out. We have updated the paper to fix the typos in citations.
>
> **Figure 1 is not referenced in the text.**
>
> Thanks for pointing out. We have added the reference to Figure 1 in the updated version of this paper.
>
> **The method assumes the existing strong method for perception and prediction, which seems like quite a large requirement. The input to GPT assumes detections and their predicted trajectories, which seems to add quite a lot of complexity (both from the training and inference standpoint).**
>
> Our method does not assume **strong** perception and prediction methods. On the contrary, our method can adapt to **any** perception and prediction methods, and here we choose the perception and prediction modules in UniAD mainly for a fair comparison with this approach.
>
> Perception and prediction are indispensable for safe motion planning in the context of autonomous driving. In most state-of-the-art approaches (e.g. UniAD, ST-P3), their motion planners take perception and prediction data as inputs. Hence, it is not a large requirement for the GPT-Driver to take perception and prediction as inputs.
>
> **Related to this, the authors don't really do an ablation study of the perception/prediction module, which would give an indication of how robust is GPT to this part of the methodology.**
>
> We have actually ablated the effects of perception and prediction in Table 5, where we use perfect and UniAD's perception and prediction results as inputs to GPT-Driver. As we mentioned in Section 4.2, when replacing the perfect perception and prediction with the learned ones, the planning performance only drops slightly, which indicates the robustness of our GPT-Driver to perception and prediction errors. We are also working on ablating more perception and prediction approaches.
>
> **In eq (2) the authors say that the input to their model is a map, yet that is not the case as they don't provide the map to the model.**
>
> What we want to present in Eq (2) is motion planning, in general, can take maps as input data. Since our approach currently doesn't rely on maps, we have revised this part for better clarification.
>
> **Later they say that their model can indeed take the map as an input, but given that they represent all inputs as a text it is far from clear how can that be done.**
>
> Maps in autonomous driving can be represented in vectorized format, where the compact representation can be expressed as a graph denoting the locations and connections of lanes. The vectorized map can be further transformed into texts describing the locations and connections of the lanes. Hence, our approach has the potential to incorporate vectorized map information.
>
> **In Section 4.2 the authors say that other approaches depend on various heterogeneous inputs "which makes their systems intricate and time-consuming", yet the proposed method also depends on the same inputs since it depends on UniAD. So the authors are not being really honest in this case.**
>
> Other approaches, such as UniAD, take compact maps (H xW grids) and occupancy (H x W x T grids), in addition to perception and prediction, as inputs to their motion planner.  Extracting those compact maps and occupancy grids makes their system intricate and time-consuming. By contrast, our approach only leverages text-based perception and prediction (instead of compact maps and occupancy grids) of UniAD as inputs to GPT-Driver. Hence, we believe our argument here is **correct** and **honest**. We have revised this part to make more clarifications.
>
> **In Section 4.3 it is unclear if the authors use fully trained UniAD for generating input strings for their method, or if they use partially trained UniAD. This should be clarified.**
>
> In Section 4.3, since we are comparing the motion planning performance, the perception and prediction in both UniAD and our approach are fully trained, and only the motion planner in UniAD and GPT-Driver are trained with 1%, 10%, 50% data. We have added more clarifications in this section.
>
> **Some sort of latency analysis should be provided, beyond just a handwavy explanation from Section 4.6. This is important for the practical application of their method and is something that the authors should explore.**
>
> We agree with this point. As we discussed in Section 4.6, the major problem here is that we are unable to obtain very accurate inference time through the OpenAI APIs. Hence the latency analysis is not feasible at this moment. We will keep track of the updates of OpenAI and see whether we could add this analysis in the future.

---

> > ### Comment · Reviewer_cPYB · 2023-11-23
> > **Thanks for the response**
> >
> > I would like to thank the authors for their detailed responses.
> > The responses do clarify many of my concerns. However, some of the bigger ones remain:
> > - The latency question is still a major disadvantage and unknown of the proposed method.
> > - When it comes to the map, while I do see how the map can be provided, it is still unclear if that would actually work well, especially given some recent evidence that LLMs have trouble with mathematical reasoning. More evaluation and evidence are needed beyond just pointing out how the map info can be provided.
> > - Also, when it comes to dependence on the existing methods, it is still true that the proposed method also inherits their downsides.
> >
> > I definitely appreciate the proposed idea and the authors' clarifications and edits. However, looking at the feedback from the other authors and the ongoing discussions it seems that the best course of action would be to revise the work, address in detail all the comments and requests, and resubmit.
> > Still, if the other reviewers agree on a positive outcome and champion the paper, I would not block such a decision.

---

> > > ### Author Response · Authors · 2023-11-23
> > > **Additional responses**
> > >
> > > Thank you so much for these further questions. We would like to use this opportunity to provide further responses.
> > >
> > > **Latency.** We admit that our system is not real-time as of now due to GPT API latency. However, if we use a Llama2 7B model, we believe it is possible to achieve real-time inference. Again, the goal of this paper is to investigate the potentials of LLMs applied to motion planning and we have some surprising discoveries in this study. We will leave efficiency optimization for future study.
> > >
> > > **Map.** In fact, a very recent work [1] extends our work and successfully leverages mapping through function calls. In addition to it, as mentioned above, an even simpler version would be transforming a vectorized map into text descriptions of locations and connections of lanes.
> > >
> > > **Dependency on the existing methods.** Again, our method can incorporate any perception modules and can tolerate certain perception errors, as verified in Table 1. We don't know what downsides you're referring to. We are happy to address any further concerns in this regard if you provide a specific example.
> > >
> > > Thank you again for providing very thoughtful comments on our submission. We are committed to address and discuss any further concerns you might have. We greatly appreciate further feedback and an increased score if we address your questions.
> > >
> > > [1] Jiageng Mao, Junjie Ye, Yuxi Qian, Marco Pavone, Yue Wang: A Language Agent for Autonomous Driving, ArXiv 2023.

---

### Public Comment · ~Bernhard_Jaeger1 · 2023-11-21
**Surprisingly low trajectory prediction errors.**

Several of the reviewers of this work were surprised by the low L2 error achieved by this method so I wanted to add a bit of context here.
Zhai et al. 2023 has shown that extrapolating the ego state yields state-of-the-art open loop L2 error on nuScenes even with a blind 2-layer MLP.

Unlike UniAD, GPT-driver takes the ego state/historical trajectory as input, see Figure 2, so low L2 errors are expected when performing optimization (but not with in-context learning, which is supported by Table 3).
I would recommend an experiment without ego states and history, like is done in Jiang et al. 2023 or closed loop experiments like the reviewers suggested, to make sure that the presented method does more than simple extrapolation of its input.

Additionally, very recent work (Mao et al. 2023) has pointed out that multiple definitions of the L2 (m) metric are used in the community, so I would recommend to explicitly mention which version was used to evaluate this method.


References:

Jiang-Tian Zhai, Ze Feng, Jinhao Du, Yongqiang Mao, Jiang-Jiang Liu, Zichang Tan, Yifu Zhang, Xiaoqing Ye, Jingdong Wang: Rethinking the Open-Loop Evaluation of End-to-End Autonomous Driving in nuScenes, ArXiv 2023.

Bo Jiang, Shaoyu Chen, Qing Xu, Bencheng Liao, Jiajie Chen, Helong Zhou, Qian Zhang, Wenyu Liu, Chang Huang, Xinggang Wang VAD: Vectorized Scene Representation for Efficient Autonomous Driving, ICCV 2023.

Jiageng Mao, Junjie Ye, Yuxi Qian, Marco Pavone, Yue Wang: A Language Agent for Autonomous Driving, ArXiv 2023.

---

> ### Author Response · Authors · 2023-11-22
> **Thanks for your feedback**
>
> Thanks for your feedback!
>
> **Several of the reviewers of this work were surprised by the low L2 error achieved by this method so I wanted to add a bit of context here. Zhai et al. 2023 has shown that extrapolating the ego state yields state-of-the-art open loop L2 error on nuScenes even with a blind 2-layer MLP.**
>
> Zhai et al. 2023 is not a peer-reviewed article. There is also data leakage in their officially released repository (see issues), making their results not that convincing. We believe the conclusions in this paper are not fully validated.
>
> **Unlike UniAD, the GPT-driver takes the ego state/historical trajectory as input, see Figure 2, so low L2 errors are expected when performing optimization (but not with in-context learning, which is supported by Table 3). I would recommend an experiment without ego states and history, like is done in Jiang et al. 2023 or closed loop experiments like the reviewers suggested, to make sure that the presented method does more than simple extrapolation of its input.**
>
> Thanks for your advice. We want to clarify this point from the following perspectives:
>
> (1) UniAD has been using tracks (historical trajectories) to generate the object's future trajectories in their prediction modules. During this process, they also generate the ego-object's future trajectory (motion planning results) in prediction and utilize this as an initial input to the motion planner. In addition, UniAD has two tricks to further boost the motion planning performance: they applied an IoU loss in learning and a post-optimization approach to further refine the trajectory. By contrast, our approach doesn't rely on these tricks.
>
> (2) There are debates about whether or not to use ego-states/historical trajectories in planning. From our understanding, these data naturally exist and are very easy to obtain in modern autonomous driving systems, there is no reason to completely abandon this information in motion planning. We still appreciate your suggestions and will conduct an experiment to remove the ego-states/historical trajectory in our GPT-Driver, as well as the closed-loop experiments.
>
> (3) Both in-context learning and finetuning in Table 3 are using the same data. The only difference is that in in-context learning, the input and desired outputs are collectively used as inputs to instruct the LLM, while in fine-tuning the desired outputs are used as targets to fine-tune the LLM. There is no post-optimization stage in fine-tuning.
>
> **Additionally, very recent work (Mao et al. 2023) has pointed out that multiple definitions of the L2 (m) metric are used in the community, so I would recommend to explicitly mention which version was used to evaluate this method.**
>
> We have also identified these differences in the evaluation metrics of different papers after this submission. In fact, the difference not only lies in L2 but also in the collision rate calculation. We will update the paper and report the respective performance in **all metrics** very soon.

---

> ### Public Comment · ~Bernhard_Jaeger1 · 2023-11-22
> **Thank you for your response.**
>
> Thank you for your response.
>
> > Zhai et al. 2023 is not a peer-reviewed article. There is also data leakage in their officially released repository (see issues), making their results not that convincing. We believe the conclusions in this paper are not fully validated.
>
> It is true that the first version had a data leakage issue.
> The authors have addressed this problem as far as I can see and uploaded a revised version to arXiv that showed that the leakage issue only had a minor impact and did not change the conclusions.
> Their results have also been independently reproduced on nuPlan open loop (which is similar to nuScenes open loop) by Dauner et al. 2023 which is peer-reviewed, so I would take these results seriously.
>
> > There are debates about whether or not to use ego-states/historical trajectories in planning...
>
> As you said, it is totally fine to use ego states as inputs to a planner.
> The problem only arises when methods using ego states are evaluated open loop.
> Extrapolating ego states is a simple heuristic that neural networks can use to obtain very good training loss (when using open loop training, as is typical).
> However, this extrapolation shortcut can lead to catastrophic failure in closed-loop evaluation where the ego states depend on the model's predictions (unlike in open loop).
> The most famous example of this in the closed-loop planning community is the inertia problem (Codevilla et al. 2019) where the neural network is extrapolating the ego velocity, leading to the car not starting to drive in closed-loop, when the velocity is already 0.
>
> Removing ego states is just a convenient workaround to still get meaningful results with open loop evaluation, but in general it is better to just do closed-loop evaluation where it is unproblematic to use ego states as input.
>
> > Both in-context learning and finetuning in Table 3 are using the same data. The only difference is that in in-context learning, the input and desired outputs are collectively used as inputs to instruct the LLM, while in fine-tuning the desired outputs are used as targets to fine-tune the LLM. There is no post-optimization stage in fine-tuning.
>
> What I meant is that pre-trained LLMs do not know about the extrapolation shortcut, so they are very unlikely to exploit it with in-context learning.
> During fine-tuning, the LLM might pick up on the shortcut and start exploiting it, which is of course not certain but a possible risk.
>
> Dauner, Daniel and Hallgarten, Marcel and Geiger, Andreas and Chitta, Kashyap: Parting with Misconceptions about Learning-based Vehicle Motion Planning. Conference on Robot Learning (CoRL), 2023.
> Felipe Codevilla, Eder Santana, Antonio M. López, Adrien Gaidon: Exploring the Limitations of Behavior Cloning for Autonomous Driving, International Conference on Computer Vision (ICCV), 2019.

---

> > ### Author Response · Authors · 2023-11-23
> > **Thanks for your response**
> >
> > **It is true that the first version had a data leakage issue. The authors have addressed this problem as far as I can see and uploaded a revised version to arXiv that showed that the leakage issue only had a minor impact and did not change the conclusions. Their results have also been independently reproduced on nuPlan open loop (which is similar to nuScenes open loop) by Dauner et al. 2023 which is peer-reviewed, so I would take these results seriously.**
> >
> > We didn't find such reproduction in Dauner et al. 2023. On the contrary, in Dauner et al. 2023, they claimed that "completely removing scene context (as in Zhai et al. 2023) is harmful, whereas a simple centerline representation of the context is sufficient for strong open-loop performance." The conclusion of Dauner et al. 2023 is actually centerline-based planning outperforms other learning-based approaches, but leveraging only the ego-states didn't perform well on the nuPlan benchmark. Also, on the nuPlan open-loop evaluation benchmark, Dauner et al. 2023 also leveraged the ego-states (see Figure 2) in their approach. Given the above facts, we disagree with the point that we should remove ego-states in the open-loop planning benchmark.

---

> ### Author Response · Authors · 2023-11-23
> **Cont'd**
>
> **As you said, it is totally fine to use ego states as inputs to a planner. ...**
>
> We agree with the point that open-loop planning and close-loop planning have some misalignments. However, we cannot agree with the two points you have mentioned (1) learning-based planners are actually extrapolating ego states (2) removing ego-states is good in open-loop motion planning.
>
> (1) learning-based planners are simply extrapolating ego states (as a shortcut)
>
> We believe the more appropriate statement should be **ego states serve as a strong prior for motion planners, however, learning-based motion planners do not merely rely on extrapolating ego states to perform motion planning.** We believe this statement has been supported by Dauner et al. 2023, where they leverage both ego-states and centerlines as inputs to a neural network, and they leverage a neural network to predict the future driving trajectory, instead of simply extrapolating ego states. Also, we didn't find any approach that simply extrapolates ego states can outperform learning-based approaches.
>
> (2) removing ego-states is good in open-loop motion planning.
>
> The major problem of removing ego-states in open-loop planning is this will make the L2 metric meaningless. L2, as we mentioned, is essentially measuring the proximity of a planned trajectory to a human driving trajectory. If we remove ego-states, we can plan arbitrary trajectories that are feasible and collision-free, but not close to a human driving trajectory, which makes L2 reflect nothing about the planning quality. By contrast, in the current setting, we're planning based on human driving history, and the L2 metric here essentially measures the imitation learning ability of motion planners.

---

> > ### Author Response · Authors · 2023-11-23
> > **Cont'd**
> >
> > **What I meant is that pre-trained LLMs do not know about the extrapolation shortcut, so they are very unlikely to exploit it with in-context learning. During fine-tuning, the LLM might pick up on the shortcut and start exploiting it, which is of course not certain but a possible risk.**
> >
> > Both in-context learning and fine-tuning we are using the same input data. We don't think the LLMs have different understandings of the data in the two strategies.

---

### Author Response · Authors · 2023-11-23
**Results of removing ego-states**

To answer the public comment about the ego state, we conduct an experiment disabling ego-states in GPT-Driver. Due to the time limit, we only fine-tuned the GPT-Driver on 10% data. Here are the results:

| L2   |      |      |      | Collision |      |      |      |
|------|------|------|------|------|------|------|------|
| 1s   | 2s   | 3s   | Avg. | 1s  | 2s   | 3s   | Avg. |
| 0.32 | 0.67 | 1.15 | 0.71 | 0.18  | 0.44 | 1.11 | 0.57 |

GPT-Driver still has very strong planning performance removing ego-states.

---

### Meta-Review · Area_Chair_U8Lh · 2023-12-06

**Metareview:**

Summary: This paper investigates if OpenAI GPT-3.5 can be a motion planner for autonomous vehicles. The paper states that its key idea is formulating motion planning as a language modeling problem. Another key idea is fine-tuning GPT with human trajectory data. The paper evaluates the proposed approach on nuScenes, testing if GPT-Driver can predict human trajectories from this data compared to end-to-end planning models.

Strengths: The idea of extracting useful information from GPT for motion planning is compelling. The idea and results of fine-tuning GPT with human trajectory data was surprising.

Weaknesses: Reviewers raise concerns about only open-loop evaluation (on static agent datasets) instead of closed-loop evaluation (e.g., nuPlan), and mention that the contribution is “on the shallower side”. They also raise concerns about the baselines being weak. Some reviewers raise the need to compare to more traditional planning approaches (e.g., MPC optimization-based trajectory planning). Since the ultimate evaluation of the proposed approach is to see if GPT-Driver can predict real human driving trajectories—and there is a duality between prediction and planning—I recommend that stronger comparisons should be run against SOTA prediction models, like MotionLM [1], which uses the same core idea (treat motion planning as language modeling) but doesn’t use ChatGPT as an interface. Another (older) open-source model that is worth comparing to, that works on nuScenes, and uses explicit map / state representations is Trajectron++ [2]. Running these baselines would show the value of using ChatGPT as an interface and show the value over traditional representations of the driving problem.

[1] Seff, Ari, et al. "MotionLM: Multi-agent motion forecasting as language modeling." Proceedings of the IEEE/CVF International Conference on Computer Vision. 2023.
[2] Salzmann, Tim, et al. "Trajectron++: Dynamically-feasible trajectory forecasting with heterogeneous data." Computer Vision–ECCV 2020: 16th European Conference, Glasgow, UK, August 23–28, 2020, Proceedings, Part XVIII 16. Springer International Publishing, 2020.

Additionally, these days the Waymo Open Motion Dataset (WOMD) is an extremely popular benchmark (see [1] for its usage), largely because it contains many more interactive, difficult driving examples. Since one of the claims of this paper is that GPT can handle tail-events better, I suggest pulling out these hard scenarios from WOMD explicitly, and analyzing the proposed approach’s performance compared to the benchmarks described above.

**Justification For Why Not Higher Score:**

After I carefully reviewed the manuscript and the author-reviewer discussion, I believe that a more detailed scientific analysis on this idea is needed (e.g., carefully studying tail-events, sensitivity of GPT-driver to perception errors compared to other models, how robust GPT-driver is to closed-loop interactions which induce covariate shift, etc.) as well as a more thorough comparison to other SOTA motion planning / forecasting models to strengthen the paper.

**Justification For Why Not Lower Score:**

N/A

---

### Decision · Program_Chairs · 2024-01-16

Reject